# A simulation of older adults' associative memory deficit using structural process interference in young adults

Yafit Oscar-Strom[1][*], Jonathan Guez[2,3]

1 Department of Psychology, Ben-Gurion University of the Negev, Beer-Sheva, Israel, 2 Department of Psychology, Achva Academic College, Arugot, Israel, 3 Beer-Sheva Mental Health Center, Faculty of Health Sciences, Ben-Gurion University of the Negev, Beer-Sheva, Israel

☯ These authors contributed equally to this work.
* yafit.oscar@gmail.com

## Abstract

Associative memory deficit underlies a part of older adults' deficient episodic memory due to the reduced ability to bind units of information. In this article we further assess the mechanism underlying this deficit, by assessing the degree to which we can model it in young adults under conditions of divided attention. We shall describe two experiments in this paper; these experiments investigate item and associative recognition in young adults under full- or divided-attention conditions. The secondary tasks employed were N-back like (NBL), which serves as a working memory updating task, and parity judgement and visuospatial (VS) tasks, which serve as non-working memory tasks. The results of both experiments show that only the NBL specifically affected associative recognition, while the other tasks affected item and associative memory to the same degree, indicating a general resource competition. These results presented a convergence of evidence for the associative deficit in older adults by modelling it in young adults.

## Introduction

There is a body of research on lifespan cognitive processes that specifically investigated memory performance in older adults to identify the causes of their memory decline. The research examined age-related differences in memory performance and types of information. Various studies suggested several explanations for older adults' memory decline: a reduction in attentional resources, processing speed and failure of inhibitory processes [1–4]. While all these explanations described a deficit in memory processing, they did not fully explain the significant episodic memory deficit in older adults [5, 6]. Some studies explained that the episodic memory age decline was because they struggle to bind together a contextual feature of the event [7, 8]. Based on this explanation, the Associative-Deficit Hypothesis (ADH) framework was proposed [6]. This framework indicates that deficiency in creating and retrieving associations between different components of an episode is the primary cause for the decline in older adults' episodic memory.

**Data Availability Statement:** All relevant data are within the manuscript and its Supporting Information files.

**Funding:** This study was supported by the Achva Academic College in the form of a grant to JG [grant number: 0.00.800.939-901].

**Competing interests:** The authors have declared that no competing interests exist.

To examine this framework, researchers used a task that differentiates between item and associative information [6, 9] and thus allowed its independent estimation. Participants were required to memorize a list of pairs of items (the basic units can be two items, an item and its context, two contextual elements, or more generally the representation of two mental codes) and then to perform either an item or an associative recognition task. In the item task, participants were required to decide whether a presented item stimulus appeared in the learning phase compared to non-presented items. In the associative task, participants were required to decide whether a presented pair of two items appeared as an identical pair in the learning phase or was a recombined pair. Many studies using this paradigm supported the ADH, indicating that older adults exhibit a specific associative memory deficit, compared to young adults, while the ability to retrieve the item itself remains intact [5, 10–13]. A simple index was created to measure the level of associative memory deficit (associate deficit index, ADI). This ADI is a calculation of the difference between item recognition and associative recognition performance. In this index, item memory was used as the baseline and no association can be created without its components. A higher ADI value represented a greater associative deficit [14, 15].

One question regarding the ADH is whether it presents a new direction in aging memory research or whether it can, as previously, be described as a case of episodic memory failure due to insufficient attentional resources to process the associative task. Indeed, there are studies [1, 16] that support the idea that a reduction in attentional resources underlies older adults' decline in episodic memory. Thus, it is reasonable to hypothesize that young adults under divided attention (DA) will be a good simulation of older adults. Yet, when testing item and associative recognition in young adults under DA, simple resource deficiency is insufficient to explain age memory decline. These studies found an additive effect in which both item and associative recognition were affected to the same degree [10–13]. These results support the ADH framework and show that it is an independent mechanism beyond general attentional resources.

While the importance of modulating older adults' cognition in young adults is clear, to expand our understanding regarding the locus of the processes involved, a simple DA paradigm is insufficient. We suggest that using secondary tasks that are known to employ a specific mechanism is the way to simulate older adults' memory decline in young adults.

Thus, the failure to simulate older adults' associative deficit in young adults might stem from the secondary tasks used in the above research [11, 17, 18]. It appears that in much of the research, the secondary tasks were continuous reaction time (CRT) tasks, involving monitoring, and shallow perceptual processes or requiring simple information that already exists in the long-term memory (LTM). Therefore, these tasks served more as a mirror for the attentional cost of the primary memory task (for example [19–21]) and were developed to minimize structural interference. Here we suggest a complementary direction using a task with a specific feature that will interfere and compete with the shared process of the primary memory task. This configuration might teach us about the processes involved in the primary task.

There is a claim in the literature that a working memory (WM) deficit is the cause of an age-related decline in cognitive abilities [22, 23]. Previous studies showed that older adults have reduced WM span compared to young adults [24, 25]. Other studies found that a deficit in the efficiency or processing component in WM results in its age-related decline [26–30]. More specifically, a few studies suggested that there might be a relation between a reduction in WM resources and associative deficit in the LTM of older adults [31, 32].

Following the above, it appears that the secondary tasks used in previous research did not involve WM updating or any other demanding process that requires WM, and therefore did not interfere with the associative binding. Indeed, one recent study [33] examined whether the

mediation of a reduction in WM resources explains the associative binding deficit in older adults. In those experiments, the researchers used secondary tasks that manipulated the amount of WM storage or processing resources by using math operations. Their results indicate that under DA conditions, participants obtained a differential decline in associative rather than item memory. They concluded that the kind of secondary task that is being used is important, and only a task that involves WM resources is good enough to simulate older adults' associative deficit in young adults. Yet, their research has some limitations. First, they used a secondary task that employed simple math calculations (addition and subtraction), which yielded the same additive effect to the full-attention condition. Only division math calculations yielded significant interaction, presenting associative deficits compared to full-attention condition. Furthermore, manipulating letter strings as a WM storage (e.g., Experiment 1) also presented partial support because the most demanding condition (4-letter strings) failed to support the prediction that gradually reducing WM storage resources elicits an associative deficit. Moreover, the secondary tasks used in that research did involve WM, but did not utilize a continuous task and left an indeterminate amount of time and attention to carry out other memory-related processes [13].

## The aim of the following research

The purpose of our study is to examine whether continuous WM secondary tasks can simulate older adults' associative deficit in young adults by creating a clear distinction between the cognitive mechanisms that the tasks required. The importance of using a continuous WM secondary task is that it requires the participants to divide their attention while performing the task. Therefore, such a task does not leave the participant enough time to switch between the tasks. The innovation of our experiments is that only a secondary task that involves WM updating will create an associative deficit in young adults. Each secondary task that will involve the relevant mechanism of binding in WM will lead to an associative deficit in young adults. For this purpose, we manipulated the secondary task in the DA condition. We used three different secondary tasks: N-back like (NBL), parity judgement and visuospatial (VS), and these were performed simultaneously with the memory encoding.

To understand better which secondary task will lead to an associative deficit, a closer observation of the cognitive aspects of the assignments is required. Our modified NBL task required the participants to hold in mind three digits at a time and compare them to each other (i.e., verify if they are all odd numbers in a sequence) and update the three-digit string whenever a new digit is presented. In the standard N-back task, a participant is asked to recall only one digit at a time, depending on the interval defined (N = 1, N = 2, etc.). Thus, our modified task can be presented as a 1+2+3-back version. This task is usually referred to as a WM updating measure and requires active maintenance of WM [34]. Since a participant needs to hold three digits at a time in WM and also to compare the presented stimulus to the previous stimuli, the steps of the experiment are dependent on each other. As a result, an association is created and therefore, this task requires a binding mechanism. In the parity task, participants were shown a random number from 1 to 9 and were asked to decide whether the number was odd or even. The decision whether a number is odd or even relies on information that already exists in LTM. In the VS task, participants were shown a digit on a quadrant of a screen and were required to decide which quadrant the digit appeard in and then press the matching response key. This kind of task requires only shallow perceptual resources. The parity and VS tasks are considered to be CRT tasks. As mentioned above, the CRT task does not involve the binding mechanism in WM but rather is a shallow perceptual process and relies on retrieval from LTM.

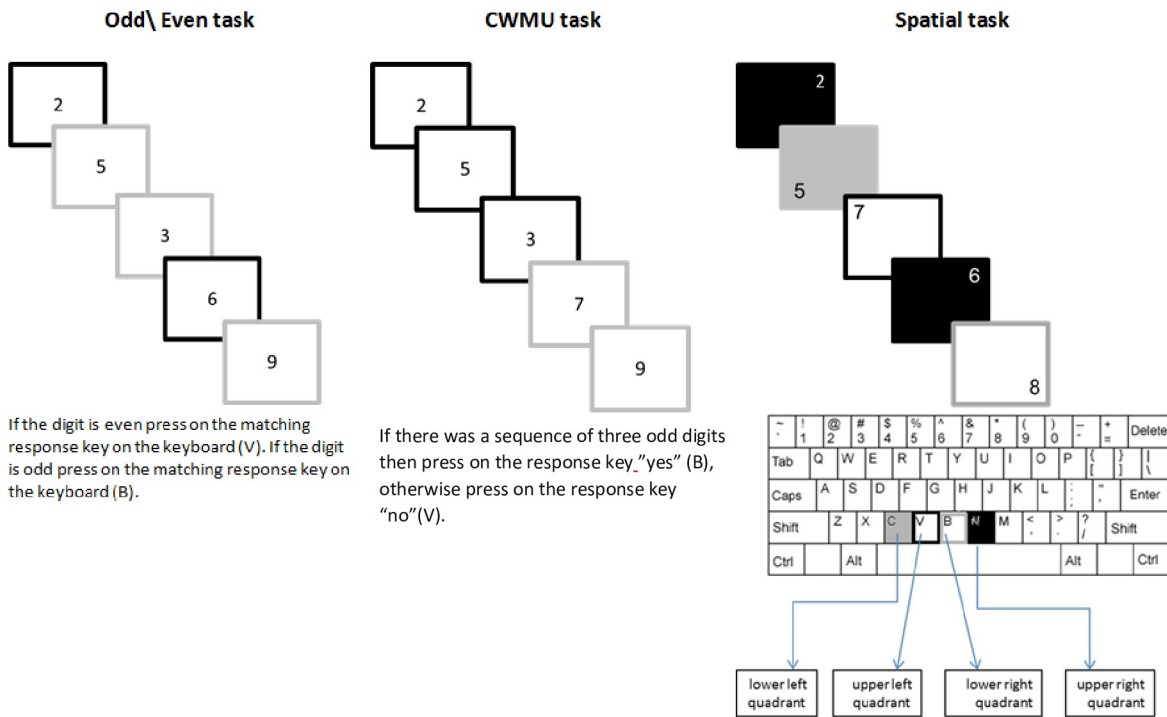

**Fig 1. Procedure of the secondary task presentation employed in Experiments 1 and 2.** CWMU = continuous working memory updating.

We claim that the NBL task engaged and competed with the mechanism needed for long-term episodic binding that leads to a recollection experience, which is essential for successful associative recognition. While recollection involves executive functioning and is associated with a clear sense of remembering, item recognition relies more on familiarity based processing [5, 35, 36]. Thus, we predict that the NBL, but not parity or VS tasks, will cause a differential deficit in young adults' memory for associative information compared to item information, although they are all attention and resource demanding.

To this end, two experiments were conducted: The first experiment compared full-attention performance for item and associative recognition to two DA conditions. The first secondary task in the DA condition was a parity judgement task, which is considered to be a CRT task, and which relies on retrieval from LTM. The second secondary task was an NBL task that is considered to be a WM task, which requires updating and therefore involves the binding mechanism. The second experiment had the same design but compared the same NBL task to a VS task. The VS task is considered to be a CRT task that relies on retrieval from LTM (see Fig 1).

## Experiment 1

### Method

**Participants.** In the current study there were 34 psychology undergraduate students (29 women) from Achva Academic College who were rewarded for their participation with course credit, an acceptable procedure in a first-year introductory psychology academic course. The mean age and education were 22.64 (1.98) years and 13.0 (0.29) years, respectively. All the participants had normal vision and hearing abilities, as was indicated in self-reports and in their

ability to report standard stimuli presented to them visually and auditorily. In addition, the participants all reported being in good health. Exclusion criteria included past/current psychiatric or neurological disorders (confirmed by the participants in a self-report) and/or a formal diagnosis of learning disabilities. The study was approved by the local institutional review board of Achva Academic College. All participants gave their written informed consent for study participation.

**Design.** Three independent variables were used: test (item/association), attention (full attention (FA)/divided attention (DA)) and task-group (NBL/parity). All factors except task-group were within subject variables. Memory accuracy was computed as the percentage of hits minus the percentage of false alarms. This measure equates the memory performance scale of the item and the associative recognition tests.

**Materials.** The stimuli were four lists of 26 pairs of words that were not related visually, auditorily or semantically. For the four lists, two versions were created, each version included all four of the lists but in one version two lists were presented in the FA condition and the other two lists were presented in the DA condition, while in the second version the list presentations were reversed. The order of the attention conditions was counterbalanced.

**Procedure.** Tasks were designed to create minimal structural interference in modality and stimuli. The memory tasks were presented auditory, and the secondary tasks were presented visually.

Before the experimental session, participants were given a practice session of the memory and the relevant secondary task to familiarize themselves with the procedure and with the secondary task.

**Memory learning phase.** The participants were tested individually in a neutral and quiet room. They heard four lists, each included 26 words pairs, at a rate of four seconds per pair, through headphones. Participants were instructed to pay attention not only to each item but also to the pairs, and were told their memory for both items and pairs would be tested.

Each list was followed by an interpolated activity of 60 sec in which the participants counted backwards in sevens from a given number, to prevent rehearsal between the learning and the test phase (recency effect). Then, after the distracting task, two memory tests, as described below (item test and association test), were presented. The order of the tests (item or associative recognition) was counterbalanced across all participants in each group, and each word appeared in only one of the tests.

**Memory test phase.** *Yes–no item recognition test*. In this test, the participants heard 16 words, one at a time, at the rate of four seconds per word. Of these, eight were target words that were selected from the studied pairs (one from each pair) and eight words were distractors. The distractors had the same characteristics as the target words, except that they had not appeared in the study phase. The participants were instructed to respond to target words with a designated "yes" response key and to press the "no" response key for a distractor word.

**Memory test phase.** *Associative recognition test*. In this test, 16 pairs of words were presented, one at a time, at the rate of four seconds per pair, by a female voice. Eight of the words were the intact pairs from the study phase, and the other eight pairs were recombined (rearranged) pairs, that is, they consisted of words taken from different study pairs (distractors). Participants were instructed to respond to target words by pressing the "yes" response key, and the "no" response key for distractor words.

**Scoring.** To assess differences between the item and associative recognition tests, we computed a measure for the percentage of hits minus the percentage of false alarms (a false alarm occurs where a non-target stimulus is identified as a target [37]) for each participant and experiment. This measure of memory accuracy equated the scales for the item and the associative recognition tests, with chance level performance at 0.00 and perfect performance at 1.00.

In accordance with this, the associative deficit is referred to as the difference between item recognition and associative recognition performance.

**DA secondary tasks.** The participants performed a secondary task during the study phase of two of the lists. This task involved a visual presentation on a computer monitor of a single-digit at the center of the screen, once every 1.5 sec, in a random order (see Fig 1).

Participants were randomly assigned to either to the NBL task-group or the parity task-group. Half of the participants performed a parity judgement secondary task, in which they were asked to judge whether the represented digit was an odd or an even number as fast as they could (by pressing a response key for "odd" or a different response key for "even"). The other half of participants received an NBL secondary task in which they were instructed to judge if there was a continuous sequence of three odd digits. In this task, participants were instructed to press on the response key "yes" whenever there was a sequence of three odd digits, otherwise the participants were instructed to press on the response key "no". While the parity task did not require the participant to maintain and update WM after each response, the NBL task required participants to keep and update their WM after each response (see Fig 1). These tasks were identical in their visual-numerical presentation and the defined pressing keys, and differed only in their instructions.

Prior to the study phase of each of the DA trials, participants were told to pay equal attention to memorizing the words and to performing the secondary digit task. Participants were told to perform the secondary task as quickly and as accurately as possible.

## Results

The mean proportion of measures for hits and false alarms for each task-group and test can be seen in Table 1. To specifically address the hypothesis tested in this experiment, we computed a three-way analysis of variance (ANOVA), with 2 (task-group) X 2 (test) X 2 (attention) for the equated measure of memory accuracy (proportion of hits minus proportion of false alarms). Results (Fig 2) indicated three significant main effects. The first main effect for attention was significant ($F(1, 32) = 20.92$, $p < 0.01$, $\eta2 = 0.39$), meaning that memory performance in the FA condition (M = .57 (0.23)) was better than in the DA condition (M = .40 (0.19)). The second main effect for test was also significant ($F(1, 32) = 43.52$, $p < 0.01$, $\eta2 = 0.57$), with performance on the item test (M = .58 (0.22)) surpassing performance on the associative test (M = .40 (0.16)). The third main effect for task-group was also significant ($F(1, 32) = 4.46$, $p < 0.01$, $\eta2 = 0.12$), with performance on the parity task (M = 0.55 (0.18)) surpassing performance on the NBL task (M = .42 (0.18)).

**Table 1. Memory performance; proportion of hits and false alarms (FAL) in full- and divided-attention conditions for the two experimental task-groups.**

| | | Parity secondary task | | | | N-Back like secondary task | | | |
| | | Full attention | | Divided attention | | Full attention | | Divided attention | |
| | | M | SD | M | SD | M | SD | M | SD |
| --- | --- | --- | --- | --- | --- | --- | --- | --- | --- |
| Item recognition | | | | | | | | | |
| | Hits | 0.77 | 0.19 | 0.76 | 0.16 | 0.75 | 0.14 | 0.63 | 0.19 |
| | FAL | 0.12 | 0.11 | 0.17 | 0.12 | 0.14 | 0.11 | 0.17 | 0.12 |
| | %Hit-%FAL | 0.65 | 0.25 | 0.58 | 0.20 | 0.61 | 0.19 | 0.47 | 0.17 |
| Associative recognition | | | | | | | | | |
| | Hits | 0.73 | 0.19 | 0.64 | 0.18 | 0.65 | 0.14 | 0.53 | 0.17 |
| | FAL | 0.15 | 0.14 | 0.22 | 0.20 | 0.18 | 0.14 | 0.38 | 0.15 |
| | %Hit-%FAL | 0.57 | 0.20 | 0.42 | 0.29 | 0.47 | 0.22 | 0.14 | 0.21 |

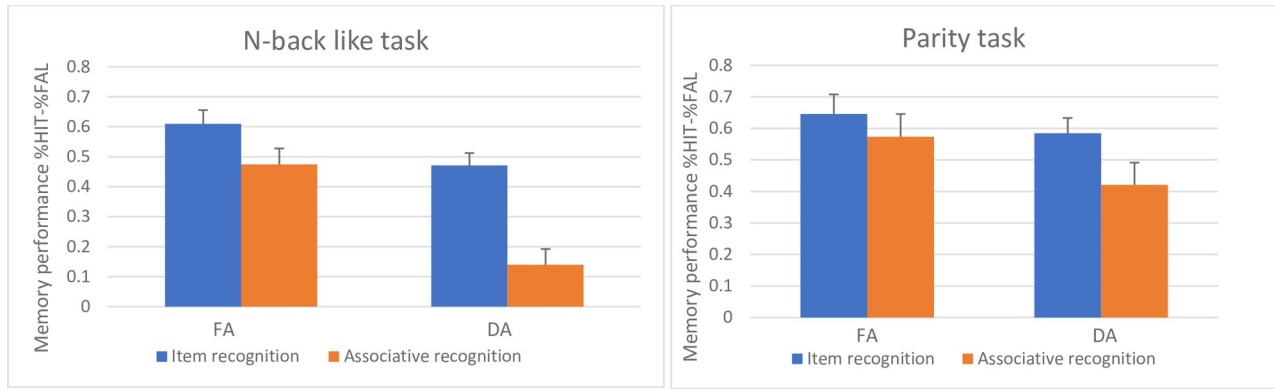

**Fig 2. Memory performance in Experiment 1; proportion of hits minus false alarms (FAL) in full- (FA) and divided-attention (DA) conditions for the two experimental groups.** The error bars represent standard error of the mean.

The interaction between attention and test was significant (F (1, 32) = 12.90, p < 0.01, η2 = 0.29). Post-hoc comparisons on the different effect of the attention condition on item and associative recognition yielded greater decrease in the DA condition between tests compared to the FA condition (F(1, 32) = 40.49, p = 0.000, η2 = 0.55; F(1, 32) = 14.13, p = 0.000, η2 = 0.30; respectively); see Fig 2B.The interaction between test and task-group was also significant (F(1, 32) = 5.09, p = 0.030, η2 = 0).14. Post-hoc comparisons showed a difference between groups only in the associative test but not in the item test (F(1, 32) = 5.99, p = 0.020, η2 = 0.16; F(1, 32) = 1.40, p = 0.244, η2 = 0.04; respectively).

The three-way interaction used to test our hypotheses did not reach significance (F(1, 32) = 2.16, p = 0.15, η2 = 0.06). Further analysis of the interaction between attention and test were significant only for the NBL task (F(1, 32) = 12.80, p < 0.01, η2 = 0.29) but not for the parity task (F(1, 32) = 2.25, p = 0.14, ns). Planned comparison on the above significant interaction for the NBL task showed greater associative deficit for the DA condition (F(1, 32) = 37.55, p < 0.01, η2 = 0.54) than for the FA condition (F(1, 32) = 11.82, p < 0.01, η2 = 0.27); see Fig 2 and Table 1. Despite the fact that the three-way interaction was not significant, we performed the planned comparisons in order to test the simple interaction between attention and test in the NBL task, as a significant ANOVA is not a pre-condition for performing focused contrasts (see Rosnow & Rosenthal [38]).

To ensure there were no differences between the task-groups in the FA condition, we performed a 2X2 ANOVA between task-group and test in the FA condition. The results indicated no significant effect for task-group (F(1, 32) = 0.72, p = 0.40, η2 = 0.01) nor for the interaction between test and task-group (F(1, 32) = 1.31, p = 0.26, η2 = 0.04). Thus, the differences between the memory performance in the DA condition stemmed from the secondary task we used.

## Discussion

The above results consistently replicate previous results. DA causes a significant decrease in memory performance [11, 12, 21, 39] and item recognition performance is superior to associative recognition performance [10–13, 39]. The advantage of these results is that they demonstrate an interaction effect between tests and attention when an NBL task was used, but not when a parity task was used. The results might suggest that a simple CRT task such as parity judgement affects item and associative recognition in an additive manner as was found in previous research, and thus presented a general resource interference. However, the NBL task,

which is considered to be a WM continuous task, presents a structural interference that affects and interacts specifically with the associative mechanism. Nevertheless, this interpretation cannot exclude other interpretations of our observations. The parity task requires less effort than the NBL task, thus it is reasonable to assume that it would increase the difficulty of the task rather than create a specific structural interference. Therefore, in the next experiment, we decided to use four choice response keys rather than two and we concentrated on this issue. Experiment 2 compares the NBL task to the VS task using the same numerical stimuli but with different instructions, which results in a separate process.

## Experiment 2

The purpose of the second experiment was to extend and replicate the findings of the first experiment by comparing the NBL task to a visuospatial task. Two groups performed memory tasks under FA and DA conditions. The tasks employed were the same as in Experiment 1, except for the VS task, where participants were instructed to indicate the spatial location of a number (see Fig 1). We hypothesize that while the VS secondary task would cause an additive effect on item and associative recognition, the NBL task would replicate Experiment 1's results by presenting interaction between tests and the attention condition.

### Method

**Participants.** In the current study there were 32 psychology undergraduate students (23 women) from Achva Academic College who were rewarded for their participation with course credit. The mean age and education were 22.58 (1.64) years and 13.33 (0.59) years, respectively. All the participants had normal vision and hearing abilities, as was indicated in self-reports and in their ability to report standard stimuli presented to them visually and auditorily. In addition, the participants all reported being in good health. Exclusion criteria included past/current psychiatric or neurological disorders (confirmed by the participants in a self-report) and/or a formal diagnosis of learning disabilities. The study was approved by the local institutional review board of Achva Academic College. All participants gave their written informed consent for study participation.

**Design & materials.** The design and materials were the same as in Experiment 1 except that spatial judgement instructions replaced the parity judgement task instructions.

*Procedure*. **The procedure was the same** as in Experiment 1. Participants were randomly assigned to either the NBL task-group or the VS task-group.

**Visuospatial task (VS).** Participants who were allocated to this task-group performed DA blocks in addition to the FA memory blocks. In the DA condition, which was performed simultaneously to the memory encoding, participants had to point to the spatial location of digits presented in a random order in four possible locations, using four matching response keys. This task involved a visual presentation on a computer monitor of single digits, one every 1.5 sec. the digit was presented on the screen in a random order in such a manner that each digit was presented in a quadrant of the screen. Participants had to decide in which quadrant the digit appeared and press on the matching response key (see Fig 1).

**NBL task.** The NBL task was the same as in Experiment 1.

**Memory tasks.** The memory task was the same as Experiment 1.

Note that the screen presentation stayed the same, but the instructions were changed for the spatial task, and the response keys were extended to four.

### Results

The mean proportion measures for hits and false alarms for each task-group and test can be seen in Table 2. In order to specifically address the hypothesis tested in this experiment, we

**Table 2. Memory performance; proportion of hits and false alarms (FAL) in full- and divided-attention conditions for the two experimental task-groups.**

| | | Visuospatial secondary task | | | | N-Back like secondary task | | | |
| --- | --- | --- | --- | --- | --- | --- | --- | --- | --- |
| | | Full attention | | Divided attention | | Full attention | | Divided attention | |
| | | M | SD | M | SD | M | SD | M | SD |
| Item recognition | | | | | | | | | |
| | Hits | 0.72 | 0.18 | 0.64 | 0.18 | 0.70 | *0.18* | 0.71 | *0.16* |
| | FAL | 0.10 | 0.11 | 0.19 | 0.16 | 0.15 | *0.13* | 0.24 | *0.13* |
| | %Hit-%FAL | 0.62 | 0.21 | 0.44 | 0.19 | 0.54 | *0.25* | 0.47 | *0.17* |
| Associative recognition | | | | | | | | | |
| | Hits | 0.64 | 0.15 | 0.57 | 0.13 | 0.65 | *0.14* | 0.50 | *0.17* |
| | FAL | 0.13 | 0.17 | 0.32 | 0.13 | 0.22 | *0.20* | 0.36 | *0.15* |
| | %Hit-%FAL | 0.51 | 0.25 | 0.27 | 0.12 | 0.43 | *0.27* | 0.14 | *0.22* |

computed a three-way ANOVA, with 2 (task-group) X 2 (test) X 2 (attention) for the equated measure of memory accuracy (proportion of hits minus proportion of false alarms). Results (Fig 3) indicated two significant main effects. The first main effect was for attention ($F(1, 30) = 23.83$, $p < .01$, $\eta2 = .44$) where memory performance in the full-attention condition ($M = .53$ (0.22)) was better than in the divided attention condition ($M = .33$ (0.14)). The second main effect was for test ($F(1, 30) = 31.83$, $p < 0.01$, $\eta2 = 0.51$) with performance on the item test ($M = .52$ (16)) surpassing performance on the associative test ($M = .34$ (0.17)).

The interaction between attention and test was also significant ($F(1, 30) = 7.01$, $p < 01$, $\eta2 = 0.19$). Post-hoc comparisons on the different effect of the attention condition on item and associative recognition showed greater decrease in the DA condition between tests compared to the FA condition ($F(1, 30) = 36.89$, $p = 0.000$, $\eta2 = 0.55$; $F(1, 30) = 6.84$, $p = 0.013$, $\eta2 = 0.18$; respectively).

As in Experiment 1, the three-way interaction was not significant ($F(1, 30) = 2.43$, $p = 0.12$, ns). Further analysis yielded that the interaction between attention and test were significant only for the NBL task-group ($F(1, 30) = 8.85$, $p < 0.01$, $\eta2 = 0.22$) but not for the VS task-group ($F < 1$). Planned comparison on the above significant interaction for the NBL task-group showed greater associative deficit for the DA condition ($F(1, 30) = 33.21$, $p < 0.01$, $\eta2 = 0.52$) than for the FA condition ($F(1, 30) = 3.58$, $p = 0.07$, ns); see Fig 3.

As in Experiment 1, in order to ensure there were no differences between the task-groups in the FA condition, we performed a 2X2 ANOVA between task-group and test in the FA

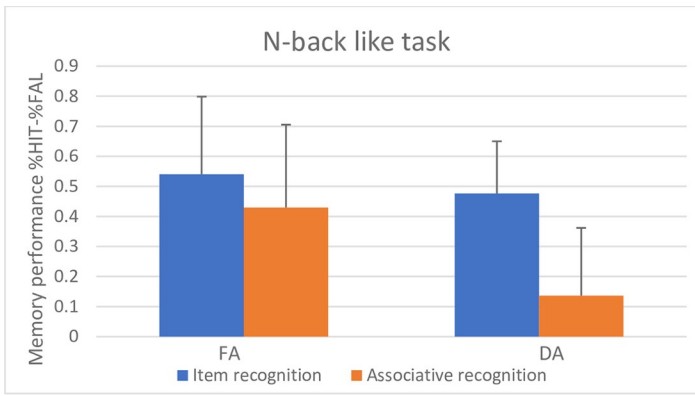 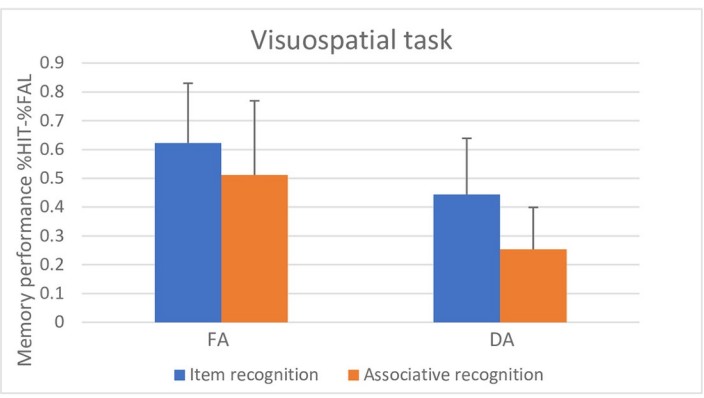

**Fig 3. Memory performance in Experiment 2; proportion of hits minus false alarms (FAL) in full- (FA) and divided-attention (DA) conditions for the two experimental groups.** The error bars represent standard error of the mean.

condition. The results indicated no significant effect for task-group (F(1, 30) = 1.10, p = 0.30, η2 = 0.03) nor for the interaction between test and task-group (F(1, 30) = 0.00, p = 0.99, η2 = 0.00). Thus, the differences between the memory performance in the DA condition stemmed from the secondary task we used.

## Discussion

Experiment 2 replicated and extended the results of Experiment 1. DA caused episodic memory deficiency, but only the NBL task made a differential effect in item and associative recognition performance compared to the FA condition.

In Experiment 1 we used a parity judgement task, but in Experiment 2 we used a VS task. The VS task compared to the parity and NBL tasks required participants to use four response keys rather than two. As such, it was more difficult, at least in terms of response production, and yet an additive effect was observed. Furthermore, taking the item recognition test as a baseline, it seems that the VS task did affect memory performance compared to the full-attention condition even more than the NBL task did (F(1, 30) = 7.23, p < 0.05, η2 = 0.19; F(1, 30) = 1.03, p = 0.32, η2 = 0.03; respectively); see Fig 3. Thus, the above result supports the specific effect of the NBL task on associative recognition.

Again, and in line with our hypothesis, only a task that involves the relevant mechanism for the creation of an association between two units of information, such as our NBL task, will lead to associative deficit.

## Further analysis across experiments

To directly compare the associative deficit across experiments and under different conditions, we used the ADI. Four attention conditions were compared in a one-way ANOVA: FA–participant under full attention; DA-parity; DA-VS; DA-N-back. Results are presented in Fig 4. The main effect of the attention condition was significant (F(3, 130) = 7.24, p = 0.00, η2 = 0.14).

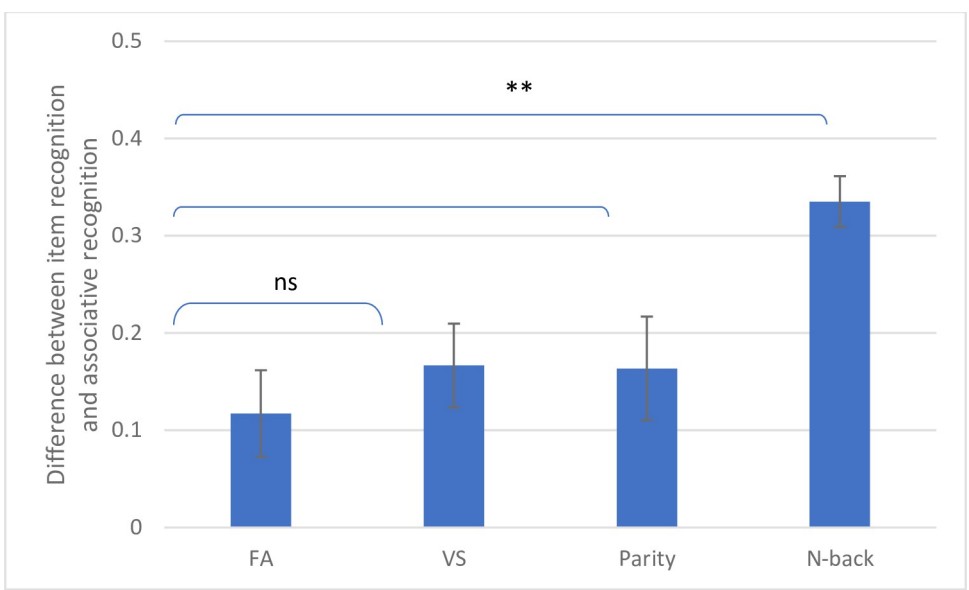

**Fig 4. Associative deficit index (ADI) in the different attention conditions across the two experiments.** FA = full attention, VS = visuopatial, **p ≤ 0.01.

Planned comparisons showed that the two CRT DA tasks (parity judgement and VS) caused the same associative deficit (F(1, 130) = 0.00; e.g., about 16% decrease from item to associative recognition), and both were not different from the associative deficit in the FA condition (F(1, 130) = 1.03, p = 0.31, η2 = 0.01; e.g., about 11% in the FA condition). All three were significantly lower than the associative deficit caused under the WM updating task (N-back) (F(1, 130) = 15.47, p = 0.00, η2 = 0.11; e.g., about 33% in the N-back DA condition). These results emphasize our hypothesis that only the N-back task simulates associative deficit in young adults. We can draw from this observation that only tasks that employed WM updating, but not a dual task that competes with attentional resources, can affect associative binding. This simulation suggests that older adults suffer from damage to the same cognitive process involved in WM updating.

## General discussion

The purpose of this research was to create process interference in young adults' episodic memory, to simulate older adults' associative deficit. This was achieved by using a continuous WM updating secondary task that involved the relevant mechanism (NBL). We suggest that only the involvement of a mechanism that is related to episodic binding (of time, objects, location, etc.) will lead to a recollection experience, which is essential for successful associative recognition. Therefore, we predicted in our task analysis that only an NBL secondary task, which is a task that requires WM updating and involves the relevant mechanism of comparison association between stimuli, would compete with associative memory binding. This, in turn, would cause a differential deficit in young adults' memory for association compared to item information, as obtained in older adults' performance. In the planned comparisons that we performed in both experiments, we found that the secondary task, whether parity judgement or VS, obtained additive deficits in the item and association recognition under the DA condition compared to the FA condition. However, in the NBL task, we found greater associative deficit for the DA condition compared to the FA condition, a pattern that characterizes the associative deficit of older adults.

Our findings replicate and expand the findings in the literature. In Experiments 1 and 2 we used the DA condition in three different secondary tasks. Two tasks out of three were CRT tasks—parity judgement and VS tasks. These kinds of tasks rely on semantic retrieval from LTM. These findings were supported by other research in this area that found an additive deficit in item and associative information [10–13]. In other words, these kinds of tasks were not good enough to simulate an associative deficit in young adults as could be observed in older adults.

Additionally, our results support and expand another study that successfully simulated older adults' associative deficit in young adults, by manipulating the amount of WM storage or processing resources [33]. Their results indicate that under DA conditions, participants obtained differential decline in associative rather than item memory. However, their findings showed only partial support for associative deficit in young adults. In our experiment, we successfully simulated older adults' associative deficit in young adults by using an NBL task, which is usually referred to as a WM updating measure, as well as active maintenance of WM. The innovation of our experiments is that only a continuous WM secondary task that requires updating, involves the relevant mechanism of binding and will lead to associative deficit in young adults.

In the above study [33], the researchers reported that associative deficit in young adults resulted from the amount of WM storage or processing. However, in another study [26] that criticized those results, it was claimed that their results showed an ordinal interaction, since

binding memory in young adults was worse in comparison to their item memory at FA, and in the DA condition it increased. In contrast, the test X DA interaction in our Experiment 2 presents a disordinal interaction, which therefore reduces such a possible claim.

One possible criticism of the above results is that the NBL task is much more demanding and thus it causes a specific associative deficit. This task-demanding hypothesis was tested in the literature and consistently presented an additive result on the test (item/association) X task-group (easy/hard) interaction [10]. To target this possible criticism directly, we moved to four response keys in a VS task in Experiment 2, compared to only two response keys in the NBL task. Furthermore, considering item performance as a baseline condition, it can be seen that comparing the deficit from full attention to divided attention, the VS task affects performance much more than the NBL task ($F_{(1, 30)} = 7.23$, $p < 0.05$, $\eta2 = .19$; $F_{(1, 30)} = 1.03$, $p = 0.31$, $\eta2 = 0.03$; respectively). This pattern was not found when comparing a parity task to an NBL task, in which the former did not affect item recognition and the latter was close to being significant ($F_{(1, 32)} = 1.11$, $p = 0.29$, $\eta2 = 0.03$; $F_{(1, 32)} = 4.14$, $p = 0.05$, $\eta2 = 0.11$; respectively). These patterns found in both experiments suggest that task demand cannot explain our results when testing its effect on a regular item recognition task.

Another alternative explanation that cannot be excluded is that both the NBL task and the creation of associations employed inhibitory processes. In the process of creating an association, one needed to bind and retrieve an A-B pair and to inhibit an A-C pair, in which all A-B-C stimuli were presented in the learning phase (unlike item recognition in which the distractors are new). In one research that targeted this hypothesis, the results showed that inhibition processes cannot simulate associative deficit and the authors concluded that associative deficit seems to be independent of other cognitive processes, including inhibitory and other resource-demanding processes [40].

## Future directions

The present research succeeded in obtaining an associative deficit in young adults by performing a secondary task that requires updating. Other than the current research, there is only one piece of evidence in the literature that showed an associative deficit in young adults by using WM storage or arithmetical calculations as a secondary task, tasks that require encoding or processing [33]. The common denominator for all the tasks is the involvement of the frontal-hippocampal circuit, and therefore more replications are needed by using different methods that involve this circuit.

Another task that can be used to simulate older adults' associative deficit in young adults is the operation span (OSPAN) task. This task is a complex WM task that requires encoding, maintenance, storage and the processing of information. When tackling this sort of task, participants were required to decide whether the presented equation is correct or not. Between the presentation of the equations, they see letters and are required to remember them in the correct order and for this, updating is required. Additionally, there is evidence in the literature that an OSPAN task involves the same brain region as a traditional neuroimaging working memory task (TNWM) like NBL. Furthermore, the OSPAN task produced robust activity in the common brain region used when carrying out the above tasks but in particular, there is greater activation of the hippocampus [41]. Thus it can be assumed that if we use the OSPAN task, greater associative deficit will be obtained than in a similar task to the NBL task.

In line with the same logic, and to examine the linear decline of associative deficit in young adults, it would be beneficial to manipulate the N-back task. EEG research in the domain of the NBL task has shown that the MTL/H (medial temporal lobe/hippocampal complex) is more activated during a 3-back task rather than a 1-back task [42]. We claim that only the

involvement of the prefrontal-hippocampal circuit during DA led to an associative deficit in young adults. Therefore, it will be interesting to examine the influence of the 3-back versus 1-back tasks on the associative deficit. We hypothesize that during a 1-back task, the associative deficit will be very small.

Previous studies that examined divided attention in young adults reported on minimum memory deficit during retrieval compared to encoding [43, 44]. Our study simulated older adults' associative deficit by using a dual task that required WM updating, only in the encoding phase. Therefore, it is important to investigate the associative deficit in young adults under DA conditions using a dual task at retrieval. Hence, future research needs to simulate older adults' associative deficit in young adults by using a dual task in the test phase in parallel to, and separated from, the encoding phase. It is possible that associative deficit will be obtained not only in the encoding but also in the retrieval phase and therefore will generate a larger associative deficit. Such results, if obtained, might provide a more accurate understanding of older adults' associative deficit.

## Conclusion

The present study simulated a pattern of associative deficit in young adults. This was achieved by using secondary tasks that involved continuous working memory updating and without CRT tasks. Our interpretation is that only the involvement of the relevant mechanisms of bindings will lead to an associative deficit in young adults, similar to that in older adults. Further studies in the neuroimaging field are required to determine if the frontal-hippocampal circuit is involved when performing an NBL task and creating the associative binding. The importance of this study is that it presents a convergence of evidence for the associative deficit in older adults by modelling it in young adults.

## Supporting information

**S1 Data. YafitOscar-NBack like vs Parity18-9-2020 (1).**
(XLS)

**S2 Data. YafitOscar-NBack like vsVS-exp2-18-9-2020 (2).**
(XLS)

## Author Contributions

**Writing – original draft:** Yafit Oscar-Strom.

**Writing – review & editing:** Jonathan Guez.

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
