## [Decision Letter · Decision Letter 0]

25 Nov 2020

PONE-D-20-30627

A simulation of older adults’ associative memory deficit

using structural process interference in young adults

PLOS ONE

Dear Dr. Oscar-Strom,

Thank you for submitting your manuscript to PLOS ONE. After careful consideration, we feel that it has merit but does not meet PLOS ONE’s publication criteria as it currently stands. In particular, the research question needs to be clearly motivated and discussed in the context of the relevant literature, and there are open questions regarding the methods and results. Therefore, we invite you to submit a substantially revised version of the manuscript that addresses the points raised during the review process.

Editor Comments:

Two expert reviewers assessed this work, and they largely agree that substantial revisions are required to make this manuscript acceptable for publication. Both reviewers agree that the introduction and discussion cover aspects (e.g. neural processes) that are speculative and irrelevant to the research question at hand, while omitting more relevant literature. These sections should thus be rewritten to reflect the relevant background, and appropriately motivate the current experiments. It is also important to clarify the choice of task, and whether the intention was to use a task (NBL) with a working memory (WM) component compared to the other tasks not involving WM.

The reviewers also comment that insufficient information is provided regarding participants and tasks.

The other comments mainly concern the statistical analyses, where additional information is required to aid interpretation of the results.

Please note that while novelty is not a publication criterion for PLOS ONE, any published work need to be methodologically sound, and the conclusions need to be supported by the data.

We look forward to receiving your revised manuscript.

Kind regards,

Maria Wimber

Academic Editor

PLOS ONE

Journal Requirements:

2. Our internal editors have looked over your manuscript and determined that it is within the scope of our Cognitive Developmental Psychology Call for Papers. The Collection will encompass a diverse range of research articles in developmental psychology, including early cognitive development, language development, atypical development, cognitive processing across the lifespan, among others, with an emphasis on transparent and reproducible reporting practices.  Additional information can be found on our announcement page: https://collections.plos.org/s/cognitive-psychology.  If you would like your manuscript to be considered for this collection, please let us know in your cover letter and we will ensure that your paper is treated as if you were responding to this call. Please note that being considered for the Collection does not require an additional peer review beyond the journal’s standard process and will not delay the publication of your manuscript if it is accepted by PLOS ONE. If you would prefer to remove your manuscript from collection consideration, please specify this in the cover letter.

Reviewers' comments:

Reviewer's Responses to Questions

**Comments to the Author**

1. Is the manuscript technically sound, and do the data support the conclusions?

Reviewer #1: No

Reviewer #2: Partly

2. Has the statistical analysis been performed appropriately and rigorously? 

Reviewer #1: Yes

Reviewer #2: Yes

3. Have the authors made all data underlying the findings in their manuscript fully available?

Reviewer #1: Yes

Reviewer #2: Yes

4. Is the manuscript presented in an intelligible fashion and written in standard English?

Reviewer #1: No

Reviewer #2: Yes

5. Review Comments to the Author

Reviewer #1: The paper proposed here explores the impact of divided attention at encoding on subsequent item and associative memory in young adults, across two experiments that vary on the task used to create divided attention (Experiment 1 using N-back like (NBL) task vs. parity decision and Experiment 2 using NBL task vs. a visuo-spatial decision task), with the aim to simulate the age-related associative memory deficit in young subjects. I have several major concerns about the paper in its current form, that make me lean towards recommending its rejection.

I found the introduction to be incomplete. The use of references is not specific enough, and references in general are not sufficiently provided given the wide literature existing on the topic of age-related associative memory decline. Some details that are irrelevant for the current study are provided (e.g., the associative index, that is then not used in the analyses, or brain regions associated with the cognitive mechanisms while the study does not include neuroimaging data), while I found that the details necessary to understand the rationale of the current study were lacking (i.e (p.2), why the previous studies manipulating divided attention to simulate the associative decline weren’t good enough to interfere with the specific associative binding process; this needs to be much more developed in my opinion).

The methods should give more information concerning the participants: was the sample size determined by a-priori power analysis? Were the participants screened for any neurological or psychiatric history? As participants were divided into two groups, were the groups matched in terms of age and education? Was the study approved by an Ethics Committee?

Concerning the methods still, I am not convinced by the choice of the secondary tasks and what they allow to show evidence for, and I did not find the information put clearly in the introduction or hypotheses. From my understanding, only the NBL task requires working memory, while the two other secondary tasks to which it is compared do not; they are parity and visuo-spatial judgment tasks. So, if I understood correctly, the task comparison implies working memory task vs. visuo-spatial or parity decision, non-working memory tasks. I think this should be put very clearly to make it clear to the reader what are the stakes of such comparison, and what they are not.

I have one problem with the results where the attention (full or divided; within subject factor) and task (NBL or alternative secondary task; across subjects factor) variables are analyzed in a factorial design. Caution should be given to indicate whether there is, or not, a difference between groups in the full attention condition (e.g., according to the attention x task interaction). Indeed, one would expect to observe no “group-task” difference in the full attention condition, as there is no secondary task to perform in that condition. Showing this absence of difference would help ensuring that the effect of task is indeed caused by the actual secondary task, when there is one, and not group difference in “baseline” (full attention) memory. Yet, these results are not reported, but there is a main effect of task, and thus, of group.

In addition, I have a great concern about the discussion part, which I found highly speculative in several aspects. Among my concerns are that many of the conclusions drawn are not supported by the data as they imply brain regions, but also are extended to other type of populations than older adults whose associative decline was simulated, such as patients with post-traumatic stress disorder.

Finally, I found that the paper would highly benefit from some English editing.

Reviewer #2: With two behavioral experiments, the present study compared item and associative memory performance for word pairs encoded with full vs. divided attention. In each experiment, participants encoded half of the stimuli with full attention, and the other half of the stimuli while also performing one of two secondary tasks (N-back like task and parity task in experiment 1, N-back like task and visuospatial task in experiment 2). Findings indicated that the N-back like task affected associative memory to a larger extent than item memory. By contrast, the other two tasks showed comparable influences on item and associative memory performance. I find these findings are interesting, although a few questions and comments need to be addressed.

1. From an overall point of view, the current study provides behavioral evidence for the shared cognitive processes recruited by NBL and associative encoding. There is a decently sized discussion of the potential neural mechanisms contributing to these processes in both the intro and discussion, which I’m not sure are clearly tied to the predictions/results, given that this is a behavioral study, and no direct comparison was made between the neural correlates of NBL and associative encoding. I would suggest the authors to focus more on the implications of the findings from a cognitive perspective.

Methods:

2. Were the participants randomly assigned to the groups with different secondary tasks?

3. Please clarify the instruction for the FA condition.

Results:

4. Across the two experiments, the group x test x attention interaction term from the initial 2 (task) x 2 (test) x 2 (attention) ANOVA was not significant. Therefore, it is not justified to examine the attention x test interaction for each group.

5. In p9, paragraph 1: the author mentioned larger associative deficit in the DA condition than the FA condition. Was the deficit measured as ADI? Please clarify the operational definition of associative deficit in the method section. Also, it would be informative to show whether the associative deficit (relative to the FA condition) was larger in the NBL task compared to the other two tasks.

6. Table 1&2: It would be helpful to add one row for each memory type to present the hits-FAs. It seems that in the FA condition, the group with NBL task in both experiments had lower associative memory performance compared to the other groups. Have the authors compared memory performance between the task groups? Is it possible that poor associative memory in the NBL group drove the significant attention x test interaction in this group?

Discussion:

7. p13, paragraph 2: Because this is a behavioral study on young adults only, it is difficult to draw a conclusion that circuit brain areas that are involved in the NBL tasks are impaired in older adults. Is there any behavioral evidence linking performance on N-back task to the performance of associative memory?

8. p14, paragraph 2: I’m not sure if the findings from PTSD and ASD patients are relevant, given that the main focus of the current study was to simulate age-related associative deficit in young adults.

9. p14, paragraph 3: Could the authors further discuss the study of Hara et al. (2015)? I think this study provide consistent findings with the current study that WM processing is a key component contributing to associative encoding.

6. PLOS authors have the option to publish the peer review history of their article (what does this mean?). If published, this will include your full peer review and any attached files.

Reviewer #1: **Yes: **Emma Delhaye

Reviewer #2: No

---

## [Author Response · Author response to Decision Letter 0]

20 Feb 2021

The manuscript has been revised. we have addressed the reviewers comments in the response to reviewers document.

---

## [Decision Letter · Decision Letter 1]

10 Mar 2021

PONE-D-20-30627R1

A simulation of older adults’ associative memory deficit

using structural process interference in young adults

PLOS ONE

Dear Dr. Oscar-Strom,

Thank you for submitting your manuscript to PLOS ONE. After careful consideration, we feel that it has merit but does not yet meet PLOS ONE’s publication criteria as it currently stands. Therefore, we invite you to submit a revised version of the manuscript that addresses the points raised during the review process.

The two original reviewers of this manuscript have now assessed the revision. As you can see from their comments, Reviewer #1 still has substantial concerns that require major revisions. Comments of Reviewer #2 are more minor in nature, but still require additional revisions. To summarize,

- Both experts agree that there are results reported in the Discussion section, which need to be moved to the Results section.

- Some theoretical arguments regarding the relationship between WM, LTM and the hypothesized binding mechanism need to be clarified.

- There are speculations about the role of MTL and mPFC in supporting memory retrieval that do not relate to the central aims and results of this study. The relationship should either be made explicity and very clear, or these sections should be removed.

- There are also further requests where methods are unclear, which need to be addressed.

- The use of the term "associative deficits" and other terms needs to be clarified and kept consistent across the manuscript.

- Finally, it is strongly recommended that the authors have their manuscript language-edited by a native English speaker to enhance intelligibility.

We look forward to receiving your revised manuscript.

Kind regards,

Maria Wimber

Academic Editor

PLOS ONE

Reviewers' comments:

Reviewer's Responses to Questions

**Comments to the Author**

1. If the authors have adequately addressed your comments raised in a previous round of review and you feel that this manuscript is now acceptable for publication, you may indicate that here to bypass the “Comments to the Author” section, enter your conflict of interest statement in the “Confidential to Editor” section, and submit your "Accept" recommendation.

Reviewer #1: (No Response)

Reviewer #2: (No Response)

2. Is the manuscript technically sound, and do the data support the conclusions?

Reviewer #1: Yes

Reviewer #2: (No Response)

3. Has the statistical analysis been performed appropriately and rigorously? 

Reviewer #1: Yes

Reviewer #2: (No Response)

4. Have the authors made all data underlying the findings in their manuscript fully available?

Reviewer #1: Yes

Reviewer #2: (No Response)

5. Is the manuscript presented in an intelligible fashion and written in standard English?

Reviewer #1: No

Reviewer #2: (No Response)

6. Review Comments to the Author

Reviewer #1: I reviewed the manuscript of the paper « A simulation of older adults’ associative memory deficit using structural process interference in young adults”. This paper intends to simulate the age-related associative memory deficit in young adults through divided attention during encoding, with the hypothesis that such deficit can be simulated specifically by a working memory task requiring memory updating, but not using other non-working memory tasks. The results support this hypothesis as the associative memory decline in young adults is specifically shown following the working memory N-back like while it is not the case following two other non-working memory tasks varying in terms of resource demands.

While I think that the study is interesting and could make a nice contribution to the literature, I have several concerns about the manuscript.

First, I really think that the manuscript would highly benefit from English editing as many sentences seem grammatically incorrect, which makes the reading laborious.

Introduction

p.6, lines 139-141: “As mentioned above, the CRT task does not involve the binding mechanism in WM but rather relies on retrieval from the LTM” (same idea also appear on p. 8, lines 168-170). Does this imply that the NBL task involves the binding mechanism in WM? And if yes, in what way exactly? Also, it is not clear to me in what way both CRT tasks rely on the retrieval from LTM. Could the authors be more precise on that point?

Methods

p. 9: it is not clear to me if there was a memory test after each list, or if the memory test took place after the 4th list. If it took place after the 4th list, it is not clear to me why each list was followed by a distracting phase. Also, did the memory test mix items from the FA and from the DA conditions?

p. 10, lines 228-231: “Of these, eight were target words which were selected from the studied pairs (1 from each pair) and eight words were distractors. The distractors were eighteen words with the same characteristics as the target words, except that they had not appeared in the study phase.” Was it eight or eighteen distractors?

Results

Please homogenize the variable names, as they are sometimes called “group” and sometimes called “task”, sometimes “task-group”.

Discussion

p. 20-21: I think results reports would fit better in the result section than in the discussion part

Reviewer #2: The authors have addressed most of my prior questions. Now I only have some minor comments for the revised manuscript.

First, please incorporate the results about ADI (in discussion) into the results section.

Second, the authors mentioned the interaction of MTL/H and mPFC in supporting memory retrieval in the discussion. Is there any evidence suggesting mPFC is involved in NBL-like working memory? Again, I think the discussion of the specific brain regions is not highly relevant to the present study.

Third, as the authors described now, associative deficit is defined as the decreased performance from item memory to associative memory, which is different from the original definition by Naveh-Benjamin (2000). If the authors stick to the current definition (which means AD is present whenever an effect of test is identified), then please make sure to use this term properly throughout the manuscript. For example, please clarify the sentence in line 614: previous associative deficit studies reported on minimum memory deficit during retrieval compared to encoding.

7. PLOS authors have the option to publish the peer review history of their article (what does this mean?). If published, this will include your full peer review and any attached files.

Reviewer #1: No

Reviewer #2: No

---

## [Author Response · Author response to Decision Letter 1]

1 May 2021

Please see our responses for the reviewers in the responses file

---

## [Decision Letter · Decision Letter 2]

25 May 2021

PONE-D-20-30627R2

A simulation of older adults’ associative memory deficit

using structural process interference in younger adults

PLOS ONE

Dear Dr. Oscar-Strom,

Thank you for submitting your manuscript to PLOS ONE. After careful consideration, we feel that it has merit but does not fully meet PLOS ONE’s publication criteria as it currently stands. Therefore, we invite you to submit a revised version of the manuscript that addresses the points raised during the review process.

The manuscript has now been re-reviewed by both original reviewers. While Reviewer #2 feels their concerns are now sufficiently addressed, Reviewer #1 still has substantial concerns regarding the readibility of the introduction, the statistics and figures reported in the results section, and parts of the discussion where some conclusions do not seem supported by the results. All of these points need to be thoroughly addressed in the next revision.

We look forward to receiving your revised manuscript.

Kind regards,

Maria Wimber

Academic Editor

PLOS ONE

Additional Editor Comments (if provided):

The manuscript has now been re-reviewed by both original reviewers. While Reviewer #2 feels their concerns are now sufficiently addressed, Reviewer #1 still has substantial concerns regarding the readibility of the introduction, the statistics and figures reported in the results section, and parts of the discussion where some conclusions do not seem supported by the results. All of these points need to be thoroughly addressed in the next revision.

Reviewers' comments:

Reviewer's Responses to Questions

**Comments to the Author**

1. If the authors have adequately addressed your comments raised in a previous round of review and you feel that this manuscript is now acceptable for publication, you may indicate that here to bypass the “Comments to the Author” section, enter your conflict of interest statement in the “Confidential to Editor” section, and submit your "Accept" recommendation.

Reviewer #1: (No Response)

Reviewer #2: (No Response)

2. Is the manuscript technically sound, and do the data support the conclusions?

Reviewer #1: Partly

Reviewer #2: (No Response)

3. Has the statistical analysis been performed appropriately and rigorously? 

Reviewer #1: No

Reviewer #2: (No Response)

4. Have the authors made all data underlying the findings in their manuscript fully available?

Reviewer #1: Yes

Reviewer #2: (No Response)

5. Is the manuscript presented in an intelligible fashion and written in standard English?

Reviewer #1: No

Reviewer #2: (No Response)

6. Review Comments to the Author

Reviewer #1: I acknowledge that the authors did a good amount of work in revising the manuscript according to the concerns that I raised. While I see an improvement in the quality of the manuscript, I still have several major concerns preventing publication.

- The introduction still needs to be substantially revised for the sake of clarity and smoothness. I still had trouble reading it, and I found the reading highly laborious. Here are a few examples of the kind of things that lack clarity, but the revision needs to be extended to the entire introduction section:

o P.2: «Various studies suggested several explanations to older adults’ memory decline: a reduction in attention, processing resources and speed [1-3], a decline in memory performance…» : the authors state that older adults’ memory decline is explained by a decline in memory performance.

o P. 4: «Failures to simulate older adults’ associative-deficit in younger adults stems from the secondary tasks used in the research [10, 16-17]» : it is not clear if this is a result shown in the literature, or is the hypothesis that the authors wish to test, or is it a hypothesis suggested by other authors?

o P. 4: «Additionally, these results support the possible validity of an independent associative mechanism» : it is not clear what results this refers to

o P. 5: «Moreover, the secondary tasks used in the this research do involve WM, but not continuous WM. Namely, each step in the experiment is independent of the other [12].»: it is not clear why and how this is important, while it should be clear at this stage of the introduction

- Result section:

o P. 11: Could the authors detail the effects of the interactions that are significant, using post-hocs? (attention x test & attention x task-group)

o P. 11: «Further analysis of the interaction between attention and test were significant only for the NBL task (F (1, 32) = 12.80, p < .01, η2 = .29) but not for the parity task (F (1, 32) = 2.25, p = 0.14, ns.)» : in my understanding, this is actually a further analysis of the triple interaction, which is non-significant

o P.16: could the authors detail the direction of the significant interaction using post-hocs? (attention x test)

o Intermediate discussions p. 13 and p.18: caution in the interpretation of the results should be taken, as the triple interactions were not significant

- Overall, in the interpretation of the results in the discussion, I think caution should be taken in sentences like, e.g., «The present research succeeded in obtaining an associative-deficit in younger adults by performing a secondary task that requires updating» as the triple interactions were non-significant in both experiments

- Figure 2 and 3: precise in the legend to which experiment each figure refers to. Also, in Figure 2, please homogenize the graphs so that we can visually compare performance across tasks

Reviewer #2: (No Response)

7. PLOS authors have the option to publish the peer review history of their article (what does this mean?). If published, this will include your full peer review and any attached files.

Reviewer #1: No

Reviewer #2: No

---

## [Author Response · Author response to Decision Letter 2]

7 Jul 2021

The responses are in the responses to reviewers file.

---

## [Editor Report · Decision Letter 3]

20 Jul 2021

PONE-D-20-30627R3

A simulation of older adults’ associative memory deficit

using structural process interference in young adults

PLOS ONE

Dear Dr. Oscar-Strom,

Thank you for submitting your manuscript to PLOS ONE. After careful consideration, we feel that it does not fully meet PLOS ONE’s publication criteria as it currently stands. Therefore, we invite you to submit a revised version of the manuscript that addresses the points raised during the review process.

This second revision of the manuscript was now reviewed by the editor, due to unavailability of the original reviewer ( Reviewer #1 who still had substantial concerns). The authors addressed most of the concerns raised by Reviewer #1, including the statistical analyses, and a restructuring and clarification of many points in the introduction. Before this study can be accepted for publication, the following revisions need to be undertaken.

(1) There are still many grammatical errors throughout the manuscript, in particular in the introduction and abstract, which should be corrected to improve the quality and readability of the manuscript. These sections should also avoid vague language like “it seems” and “it appears”, and instead clearly state what can be inferred based on the existing literature.

(2) The figure legends should be expanded to specify exactly what is represented by the bar graphs and error bars (e.g., mean +/- SEM). They should also contain an explanation of the abbreviations (FA and DA) in the legends themselves. Finally, text in the figures should be corrected for typos (e.g., “visuospatial” instead of “visuospatial” in Fig. 3.

We look forward to receiving your revised manuscript.

Kind regards,

Maria Wimber

Academic Editor

PLOS ONE

Journal Requirements:

Additional Editor Comments (if provided):

(see above)

---

## [Author Response · Author response to Decision Letter 3]

28 Sep 2021

attached a response to reviewers file

---

## [Editor Report · Decision Letter 4]

1 Oct 2021

A simulation of older adults’ associative memory deficit

using structural process interference in young adults

PONE-D-20-30627R4

Dear Dr. Oscar-Strom,

We’re pleased to inform you that your manuscript has been judged scientifically suitable for publication and will be formally accepted for publication once it meets all outstanding technical requirements.

Kind regards,

Maria Wimber

Academic Editor

PLOS ONE

Additional Editor Comments (optional): The remaining editorial comments have been addressed.

Reviewers' comments: There were no further comments from the reviewers.

---

## [Editor Report · Acceptance letter]

26 Oct 2021

PONE-D-20-30627R4 

A simulation of older adults’ associative memory deficit using structural process interference in young adults 

Dear Dr. Oscar-Strom:

I'm pleased to inform you that your manuscript has been deemed suitable for publication in PLOS ONE. Congratulations! Your manuscript is now with our production department. 

Kind regards, 

on behalf of

Dr. Maria Wimber 

Academic Editor

PLOS ONE